# Diagnostic SARS-CoV-2 Cycle Threshold Value Predicts Disease Severity, Survival, and Six-Month Sequelae in COVID-19 Symptomatic Patients

**DOI:** 10.3390/v13020281

**Published:** 2021-02-11

**Authors:** Mattia Trunfio, Francesco Venuti, Francesca Alladio, Bianca Maria Longo, Elisa Burdino, Francesco Cerutti, Valeria Ghisetti, Roberto Bertucci, Carlo Picco, Stefano Bonora, Giovanni Di Perri, Andrea Calcagno

**Affiliations:** 1Unit of Infectious Diseases, Department of Medical Sciences, University of Torino at the Amedeo di Savoia Hospital, Corso Svizzera 164, 10149 Torino, Italy; francescovenuti1993@gmail.com (F.V.); francesca.alladio@gmail.com (F.A.); bianca.longo23@gmail.com (B.M.L.); bertucci54@yahoo.it (R.B.); stefano.bonora@unito.it (S.B.); giovanni.diperri@unito.it (G.D.P.); andrea.calcagno@unito.it (A.C.); 2Microbiology and Molecular Biology Laboratory, Amedeo di Savoia Hospital, ASL Città di Torino, Corso Svizzera 164, 10149 Torino, Italy; elisa.burdino@libero.it (E.B.); francesco.cerutti@aslcittaditorino.it (F.C.); valeria.ghisetti@gmail.com (V.G.); 3Regional Department for Infectious Diseases and Emergency DIRMEI, ASL Città di Torino, Via S. Secondo 29, 10128 Torino, Italy; carlo.picco@aslcittaditorino.it

**Keywords:** COVID-19, viral load, sequelae, outcomes, cycle threshold, severity, mortality, SARS-CoV-2 swab, predictive biomarker

## Abstract

To date, there is no severe acute respiratory syndrome coronavirus 2-(SARS-CoV-2)-specific prognostic biomarker available. We assessed whether SARS-CoV-2 cycle threshold (Ct) value at diagnosis could predict novel CoronaVirus Disease 2019 (COVID-19) severity, clinical manifestations, and six-month sequelae. Hospitalized and outpatient cases were randomly sampled from the diagnoses of March 2020 and data collected at 6 months by interview and from the regional database for COVID-19 emergency. Patients were stratified according to their RNA-dependent-RNA-polymerase Ct in the nasopharyngeal swab at diagnosis as follows: Group A ≤ 20.0, 20.0 < group B ≤ 28.0, and Group C > 28.0. Disease severity was classified according to a composite scale evaluating hospital admission, worst oxygen support required, and survival. Two hundred patients were included, 27.5% in Groups A and B both, 45.0% in Group C; 90% of patients were symptomatic and 63.7% were hospitalized. The median time from COVID-19 onset to swab collection was five days. Lethality, disease severity, type, and number of signs and symptoms, as well as six-month sequelae distributed inversely among the groups with respect to SARS-CoV-2 Ct. After controlling for confounding, SARS-CoV-2 Ct at diagnosis was still associated with COVID-19-related death (*p* = 0.023), disease severity (*p* = 0.023), number of signs and symptoms (*p* < 0.01), and presence of six-month sequelae (*p* < 0.01). Early quantification of SARS-CoV-2 may be a useful predictive marker to inform differential strategies of clinical management and resource allocation.

## 1. Introduction

Most cases of severe acute respiratory syndrome coronavirus 2 (SARS-CoV-2) infection are asymptomatic or experience self-limiting flu-like manifestations [1]. The risk of developing severe novel CoronaVirus Disease 2019 (COVID-19) is known to be associated with several characterized individual conditions [2], although additional still undisclosed factors are likely to play a role as severe infections are also seen when no comorbidities are present. In order to properly manage patients presenting with a newly discovered positive swab, several prognostic scores are under evaluation to predict in-hospital death and to discriminate between patients requiring hospital admission or not [3,4]. However, these scores rely upon variables that are extremely dependent on the timing of the evaluation along COVID-19 course, with the potential of sudden changes in a few hours. While similar scores have proven validity in acute infectious diseases presenting with full-blown illness, the same parameters might be misleading in dealing with the initial phase of the SARS-CoV-2 infection. A triphasic progressive pattern has been described in infected patients evolving toward severe clinical pictures [5], and an early assessment might not rule out the subsequent worsening of the disease. Consequently, reliance upon indicators providing better prognostic predictions would be critical in order to properly select patients requiring hospital admission, especially when exponentially increasing numbers of infections occur in a short time interval and hospitals become crowded.

In contrast to other viral infections such as HIV, HBV, and CMV, so far, there are no pathogen-specific prognostic biomarkers readily available for SARS-CoV-2. Further to the recognized risk factors for severity, the initial prognostic workup of persons infected by SARS-CoV-2 would also benefit from viral biomarkers able to predict COVID-19 evolution. In this regard, it is a current matter of debate whether SARS-CoV-2 viral load is an impactful factor in determining disease outcomes [6,7,8,9,10,11,12,13,14,15,16,17]. Previous evidence from SARS-CoV and influenza suggests that the higher the initial viral load, the worse the clinical evolution [18,19]. To date, few studies have investigated the relationships among SARS-CoV-2 viral load (usually measured by the proxy PCR cycle threshold value, Ct), and mortality, disease progression, and overall severity [7,8,9,10,11,12,13,14,15,17]. Current data point towards a plausible positive correlation between the amount of detected virus and the degree of SARS-CoV-2 pneumonia severity, hypoxemia intensity, risk of death, as well as of hematological, biochemical, and inflammatory alterations [6,9,10,11,12,17,20,21]. However, heterogeneous recruitment criteria have so far hampered reaching a final, firm conclusion on the relationship between initial nasopharyngeal viral load and individual prognosis. Lastly, the first data on the six-month follow-up from the overt disease have recently been describing a worryingly large prevalence of patients recovering with significant sequelae [22]; early biomarkers are required to stratify this large number of patients in terms of differential follow-up.

Hence, the aim of this study was to assess whether SARS-CoV-2 Ct at diagnosis may predict COVID-19 severity, related clinical outcomes, and six-month sequelae. The study was performed in a representative sample of hospitalized and outpatient COVID-19 cases whose infection was diagnosed in March and data were collected at a median follow-up time of 6 months by phone interviews and from the regional database for COVID-19 emergency, which collects data of the entire Piedmont (Italy). Our findings suggest a potential application of early SARS-CoV-2 quantification in nasopharyngeal swab to predict both short- and long-term consequences of COVID-19 in hospitalized and outpatient symptomatic cases.

## 2. Materials and Methods

We performed a retrospective cross-sectional study on data from patients with a SARS-CoV-2-positive diagnostic nasopharyngeal swab analyzed by our regional reference Laboratory (Amedeo di Savoia Hospital, Turin, Italy) in March 2020, when diagnostic samples from suspected cases from the entire region (Torino and surrounding counties, districts, and cities of Piedmont) were mainly collected by our laboratory.

Patients were stratified according to diagnostic Ct values detected from the first swab that led to a SARS-CoV-2 infection diagnosis into the following three groups: Ct ≤ 20.0, group A; 20.0 < Ct ≤ 28.0, Group B; Ct > 28.0, Group C. The 28.0 cutoff was chosen due to the 100% detection rate of SARS-CoV-2 rapid antigen in samples with a Ct ≤ 28, previously described by our laboratory [23]. Patients with a Ct ≤ 28.0 showed a higher viral load and likely a still replicating virus as testified by viral antigen expression [23]. A further stratification within this group was made according to the second cutoff of 20.0, to divide the subjects according to the amount of viral load and potential infectivity, as suggested by preliminary results of studies transfecting cell cultures by diagnostic specimens [24,25,26].

The swabs were processed using the 2019 Novel Coronavirus Real-Time Multiplex PCR kit (Liferiver Bio-Tech, San Diego, CA, USA), which targets the following three SARS-CoV-2 specific genes: RNA-dependent RNA polymerase (RdRp), nucleocapsid, and envelope. For the purpose of the study, only RdRp Ct values were considered to have one uniform proxy of viral load, since RdRp was the most specific gene among the three. The ABI Prism 7500 thermal cycler (Thermo Fisher Scientific, Waltham, MA, USA) was used for PCR amplification.

In March 2020, 1995 sample results were SARS-CoV-2-positive at our laboratory, of which 1138 swabs had a Ct value greater than 28.0 and 857 less or equal to 28. The present study was primarily designed to assess whether there was a difference in mortality between cases stratified according to Ct value at COVID-19 diagnosis. Therefore, to detect as significant an estimated difference in mortality of at least 0.2 [7], with a two-sided confidence level of 0.05, a power of 95% and a ratio between Ct values higher and lower than 28.0 of 1.3, an overall sample size of 225 was required.

Patients were randomly sampled from the frame represented by the 1995 SARS-CoV-2-positive swabs through probability sampling (random lottery extraction). The sampled individuals were reached in August–September 2020 for a telephone survey addressing COVID-19 related clinical and demographic characteristics with both the interviewed and their household contacts (an English translation of the survey is shown in Appendix A). The surveyed data were crosschecked and completed by data extrapolated using the Piedmont platform (RUPCOVID), an on-line regional database built for SARS-CoV-2 contact tracing, notification (swab results and dates), and clinical data collection (demographics, signs and symptoms at onset and at diagnostic swab, date of symptoms onset, and comorbidities).

Disease severity was classified according to a six-degree scale as follows: no hospital admission, hospitalization without oxygen support, hospitalization with support from low-flow wall oxygen to reservoir mask, hospitalization requiring continuous positive airways pressure (CPAP) support, hospitalization with intubation, and death. Signs and symptoms were clustered according to the following four main groupings: fever, asthenia, malaise, and arthromyalgia as inflammatory systemic involvement; headache, olfactory, and gustatory dysfunction as neurological involvement; nausea, vomiting, and diarrhea as gastrointestinal involvement; dyspnea, runny nose, cough, and pharyngitis as respiratory involvement.

Informed consent was obtained from all subjects involved in the study when admitted to hospitals. The consent included eventual posthumous anonymized medical data utilization (all deaths occurred during hospitalization). For outpatients never admitted to hospitals, consent was asked during the phone survey; those not consenting to the survey were discarded and their data not collected from the RUPCOVID. The study was conducted according to the guidelines of the Declaration of Helsinki and approved by the Inter-departments Ethics Committee A.O.U. Città della Salute e della Scienza, A.O. Ordine Mauriziano di Torino, and A.S.L. Città di Torino (Torino, Italy, protocol number 0065839-00304/2020, approval date 09 July 2020).

Data were analyzed through nonparametric tests (Mann–Whitney, chi-square for trend, Kruskal–Wallis, and Fisher exact tests). Post hoc analyses were performed after three-group comparisons and Bonferroni correction was applied to those yielding a *p*-value < 0.05. Eta-squared effect size for Kruskal–Wallis tests that yielded a *p*-value < 0.05 was also reported to evaluate the magnitude of the difference (the effect is deemed as small if η^2^ < 0.06, moderate if 0.06 ≤ η^2^ < 0.14, large if η^2^ ≥ 0.14). Variables with relevant biological significance or showing univariate *p* ≤ 0.10 were included in the multiple linear or ordinal logistic regressions (entry method). Categorical variables are presented as absolute values (proportion) while continuous variables as medians (interquartile range). Data analysis was performed through SPSS 25.0 (IBM stat.).

## 3. Results

### 3.1. Population

In total, 230 patients were sampled to undertake the survey. Thirty (13.0%) patients refused or never answered, among which 27 patients belonged to Group C and 3 patients belonged to Group B. The two-hundred participants included 168 survivors (194 (181–198) days after COVID-19 onset) and 32 deceased. The median age was 56 years (43–69), 116 (58.0%) were male and 188 were of European ancestry (94.0%). The distribution among SARS-CoV-2 Ct groups was as follows: 55 (27.5%) patients in Group A, 55 (27.5%) patients in Group B, and 90 (45.0%) patients in Group C. Participants’ clinical characteristics are shown in Table 1.

As shown in Table 2, the groups differed in terms of age (Group A, 64 years (39–78); Group B, 57 years (50–67); Group C, 52 years (40–63); *p* = 0.017) and time from COVID-19 onset to diagnostic swab collection (Group A, 3 days (2–5); Group B, 5 days (3–10); Group C, 5 days (3–10); *p* = 0.011). More participants in Group A presented at least one comorbidity (Group A, 72.7%; Group B, 58.2%; Group C, 45.5%; *p* = 0.006). No difference was observed in specific comorbidities with the exception of active tobacco use (Group A, 21.8%; Group B, 3.6%; Group C, 10.0%; *p* = 0.010) and hypertension (Group A, 49.1%; Group B, 23.6%; Group C, 17.8%; *p* < 0.0005). The linear Ct values inversely correlated with the number of comorbidities per patient even after adjusting for time from COVID-19 onset to swab collection (β-0.21, *p* = 0.004, see Appendix A).

### 3.2. COVID-19 Outcomes According to SARS-CoV-2 Ct

Patients requiring hospitalization were observed more commonly in Group A as compared with Group C (74.5% vs. 56.7%, *p* = 0.031, Table 2). COVID-19 severity resulted significantly worse in Group A as compared with either Groups B or C. An inverse distribution in the five categories of disease severity was observed with respect to Ct (*p* = 0.004, Table 2). Lastly, COVID-19-related six-month outcomes were worse in Group A as compared with the other groups, i.e., 29.1% of patients in Group A completely recovered at 6 months versus 70.9% and 80.0% in Groups B and C, respectively. Additionally, lethality was higher in Group A (36.4%) as compared with the other groups (Group B 12.7% and Group C 5.6%, Table 2).

At multivariate analysis (after adjusting for sex, time from disease onset to swab collection, and worst oxygen support required) lower SARS-CoV-2 Ct values together with older age and higher number of comorbidities were independently associated with higher risk of COVID-19-related death (binary logistic regression *p* < 0.0005, Table 3). Similarly, lower SARS-CoV-2 Ct values, older age, male sex, and number of comorbidities independently predicted a more severe COVID-19 in a model also including the time from disease onset to swab collection (ordinal logistic regression *p* = 0.004, Table 3).

### 3.3. Clinical Presentation at Diagnosis According to SARS-CoV-2 Ct

Due to the limited laboratory capacity at the epidemic onset, most tests were carried out on symptomatic patients and only occasionally on asymptomatic patients. Thus, the latter were a minority and no difference was observed in their prevalence (Table 1).

Ct groups differed in terms of systemic inflammation (*p* = 0.035, η^2^ 0.033), gastrointestinal (*p* = 0.041, η^2^ 0.032), and respiratory manifestations (*p* = 0.060, η^2^ 0.028) (Figure 1). Specifically, Group A presented a higher prevalence of inflammatory systemic signs and symptoms and respiratory involvement as compared with Group C and a higher prevalence of gastrointestinal and respiratory involvement as compared with Group B (Figure 1). Detailed prevalence and comparisons of single signs and symptoms are shown in Figure 1.

Linear Ct values inversely correlated with the number of signs and symptoms reported at the diagnostic swab (ρ-0.23, *p* = 0.001, see Appendix A).

At multivariate analysis (after adjusting for age, sex, number of comorbidities, and time from COVID-19 onset to swab collection) lower SARS-CoV-2 Ct values were independently associated with a higher number of signs and symptoms (linear regression *p* = 0.022, Table 3).

### 3.4. Six-Month Sequelae According to SARS-CoV-2 Ct

Lastly, we evaluated whether diagnostic SARS-CoV-2 Ct values could predict sequelae among survivors. Patients undergoing a complete recovery were significantly more frequent in Groups B and C as compared with Group A, while those still suffering from COVID-19-related sequelae were about three times more frequent in Group A than in Groups B or C (Table 2).

At multivariate analysis, lower SARS-CoV-2 diagnostic Ct values independently associated with a higher prevalence of six-month sequelae among COVID-19 survivors after adjusting for age, sex, number of comorbidities, worst oxygen support required, and time from disease onset to swab collection (binary logistic regression *p* = 0.021, Table 3).

## 4. Discussion

Among symptomatic hospitalized and outpatient COVID-19 cases at the beginning of the pandemic in Italy, we observed that disease severity, death, six-month sequelae, and the number of signs and symptoms at diagnosis distributed according to the amount of nasopharyngeal SARS-CoV-2 detected within the first week from disease onset, independently from other known determinants of COVID-19 severity.

For other viral illnesses, the initial viral load has been associated with disease severity [18,19], however, a consensus has not been reached regarding COVID-19. To the best of our knowledge, 11 studies have shown that nasopharyngeal SARS-CoV-2 Ct are not associated with, or predict, COVID-19 severity [13,14,15,20,27,28,29,30,31,32,33]. However, only six of them analyzed samples with >100 cases, the majority included only hospitalized patients, and the follow-up was relatively short. Significantly, only a minority took into account the time between COVID-19 onset and the time of swab collection.

Considering viral kinetics and immunopathology dynamics [5,28,30,34], SARS-CoV-2 viral load should be adjusted for this temporal variable to avoid misclassification of patients; indeed, this factor, together with a sample size underpowered for specific secondary outcomes, may also explain why no hypothesized differences have been observed among certain univariate comparisons between group A and B. Nevertheless, the linearity of Ct value at multivariate analyses confirmed our primary hypotheses. Further studies addressing the best, standardized, and most reliable assay-based and gene-based Ct value cutoffs should be performed to support a potential clinical and easily interpretable application of Ct values in routine practice.

If it is not a matter of snapshot issues, it could be a matter of grey scale. Some of the studies supporting no difference in disease severity according to viral load did not include the whole spectrum of COVID-19 and compared either asymptomatic with very mildly symptomatic patients or not hospitalized with hospitalized patients, with the latter ranging from mild to critical symptoms [14,15].

In contrast, studies where a positive association between nasopharyngeal SARS-CoV-2 Ct values and COVID-19 severity or outcomes was detailed are flourishing [6,7,9,10,11,12,17,34,35]. Most of them included samples of more than 100 individuals with the reference swab represented by the diagnostic one, despite only two reported the time between disease onset and swab collection [7,17]. Compared to previous studies, this study has the longest follow-up and included non-hospitalized patients. Furthermore, our retrospective design at 6 months from COVID-19 onset and the availability of data on the time between disease onset and swab collection allowed us to adjust for this factor, as well as to analyze outcomes and parameters of the infection at a time when its evolution was surely over.

Lower Ct values were associated with more signs and symptoms at diagnosis and a more frequent pattern of respiratory and systemic complaints. Our findings are in line with those recently published by others [8,12,36]. As for the latter [36], we did observe higher viral load in febrile patients as compared with afebrile patients; a greater burden of virus could induce higher inflammatory response primarily manifesting as fatigue, malaise, fever and headache. In accordance with this, plasma and nasopharyngeal SARS-CoV-2 RNA has been correlated with proinflammatory cytokines and inflammation biomarkers levels, such as IL-6 and CRP/serum amyloid A ratio [20,21].

On the contrary, we did not observe any association between nasopharyngeal viral load and prevalence of olfactory/taste disorder as described by others and plausibly explained by a higher local amount of virus and related inflammation [36]. Nevertheless, we did not quantify the dysfunction with an objective scale, since our study was underpowered to specifically assess this outcome and others have also failed to find such an association due to the several different mechanisms underlying these symptoms and potentially not all correlating with local viral replication [37]. Olfactory and gustatory dysfunction represented the second most common sequelae (7.7%), still present 6 months from the infection, after dyspnea and just before chronic cough. Other studies reported persistent olfactory and gustatory dysfunction at 6 weeks and at 3 months in the 28.2–16.7% (smell-taste loss) and 10.3% of individuals [37,38]. Up to 6 months, the dysfunction may persist in a small proportion and, together with other sequelae, was more common in patients with lower Ct values regardless of age, gender, comorbidities, and infection severity.

Recently, data on long-term sequelae among hospitalized COVID-19 cases have depicted a worrying scenario, reporting up to 76% prevalence of at least one symptom at 6 months [22]. Besides not having primarily designed the study to address all the potential sequelae of the infection with tailored examination at the end of follow-up, our lower prevalence (20.5%) can also be explained by having included asymptomatic and not hospitalized patients. A larger cohort should address the issue to tailor differential follow-up strategies according to the risk of sequelae in different subgroups of patients and potentially including the initial amount of virus as one of the factors to be considered when assessing the risk.

Our study relied on data collected by surveys and electronic medical records of hospital admitted patients but also outpatient cases; thus, it could be less controlled as compared with studies focusing on inpatient cases only, lacking data on treatments, and not further differentiating the category of patients requiring from low-flow oxygen to reservoir mask. However, in March, most of the potentially impactful treatments were not available, while recall bias may have affected signs and symptoms reliability, but less likely than that of the other outcomes. Furthermore, the heterogeneity in the clinical management of patients coming from such a large area of northwestern Italy may limit the homogeneity of certain subgroups, since oxygen administration, as well as hospitalization, varied and relied on several factors that we were not able to consider (such as physician evaluations, local guidelines, and availability of intensive care unit beds).

As we sampled only 10% of asymptomatic infections, further studies are warranted to better clarify why the relationship between Ct values (or the amount of nasopharyngeal virus) and clinical outcomes seems to follow a different pattern in subjects developing asymptomatic SARS-CoV-2 infection. Indeed, several cohorts described no difference in Ct values between symptomatic and asymptomatic patients [6,39,40], therefore, genetic, immunological, analytical, or viral factors may interpose in the relationship we have observed. Lastly, we had 13% non-responder to the survey that partially unbalanced the representativeness of our sample as compared with the Ct distribution of the sampling frame.

Currently, the emerging evidence is that early Ct values from a nasopharyngeal swab correlate with disease susceptibility (age and comorbidities, such as smoking and hypertension, both associate with potential differential expression of ACE2 receptor [41,42]) and clinical presentations and predict disease severity, survival, and sequelae in symptomatic patients.

Whether a better definition of these patterns could help at triaging newly diagnosed cases is pending, as well as potential application of SARS-CoV-2 viral load in routine clinical practice. Reporting predictive virological parameters, as already done for other infections, could inform clinician management of strategies for monitoring and allocating resources; this may be especially useful during the climax of SARS-CoV-2 waves when hospital beds and resources are limited and require programmed and evidence-based selections and timely risk stratifications.

The interpretation of a single Ct value should still be performed cautiously as it may be affected by sample collection, analyzed gene, adopted assay, and analytic limits [43]; therefore, reference Ct cutoffs should be validated and standardized by genes and assays before any application in clinical practice. Indeed, quantitative assays based on RT-PCR and other techniques are under development for exactly quantifying SARS-CoV-2 RNA and will soon be introduced in routine clinical management, integrating qualitative tests by overcoming many of these limitations. Longitudinal studies should also evaluate further viral dynamics and kinetics to better interpret Ct values in accordance with the moment of swab collection along COVID-19 course. Interestingly, it has been recently described how the variation in time of sequential Ct values from nasopharynx of infected subjects sensibly predicts the changes in clinical status of COVID-19 cases [34]. In this regard, the debate on efficacy and effectiveness of drugs with potential impact upon SARS-CoV-2 replication (such as baricitinib, low-molecular-weight heparins, and remdesivir [44,45,46]) may find answers by analyzing categories of patients stratified by Ct values and their variations to identify those where the pharmacological impact could be more evident and beneficial in terms of clinical outcomes.

## 5. Conclusions

Among symptomatic hospitalized and not hospitalized patients, we demonstrated an association of the Ct value detected in nasopharyngeal swabs collected within the first week from COVID-19 onset with COVID-19-related deaths, disease severity, and number of signs and symptoms at diagnosis as well as, to the best of our knowledge, for the first time, with the persistence of sequelae at 6 months. These relationships were retained even after adjusting for other relevant parameters already expected to affect the amount of virus at the beginning of the infection. Further confirmation of our observations and the identification of reliable standardized Ct cutoffs, together with other previously well-known determinants of COVID-19 severity, could lead to timely differentiated paths in terms of clinical monitoring and management of patients when accessing health facilities as well as tailored follow-up in terms of duration and type of required assessments. Whenever possible, randomized controlled trials and other studies that aim at assessing efficacy or effectiveness of drugs in COVID-19 should also report on the amount of initial virus or its proxy to correct for another parameter that is emerging as relevant in the outcomes of SARS-CoV-2 infection.

## Figures and Tables

**Figure 1 viruses-13-00281-f001:**
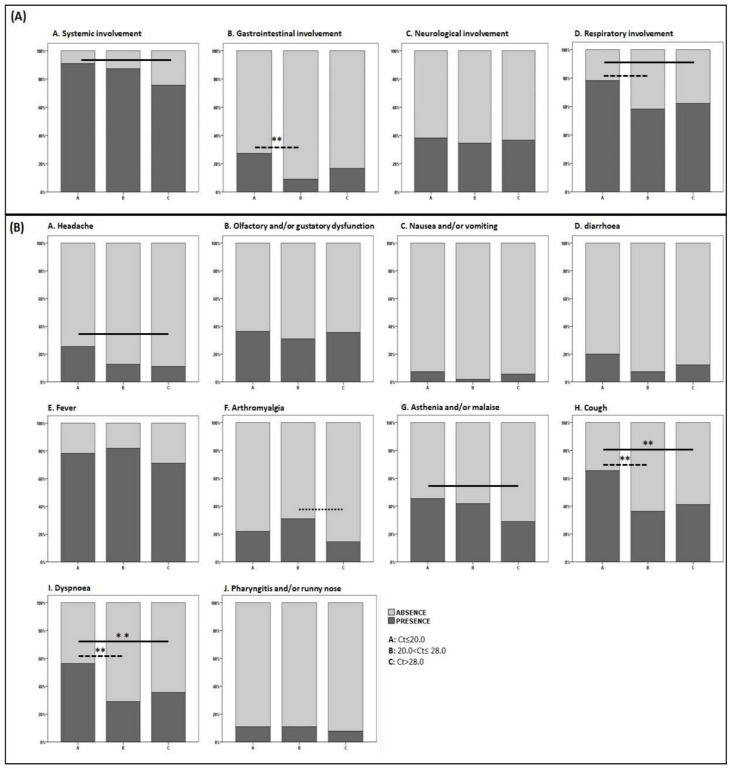
Signs and symptoms of COVID-19 at diagnosis, individual and pooled for main categories according to SARS-CoV-2 Cycle threshold. Only significant differences (Mann–Whitney) are shown as bars in the figure. Pooled signs and symptoms. (Panel **A**) (**A**) Group A 50 (90.9%) vs. Group C 68 (75.5%), *p* = 0.022; (**B**) Group A 15 (27.3%) vs. Group B 5 (9.1%), *p* = 0.014 **; (**D**) Group A 43 (78.2%) vs. Group B 32 (58.2%), *p* = 0.025 and Group A 43 (78.2%) vs. Group C 56 (62.2%), *p* = 0.046. Individual signs and symptoms. (Panel **B**) (**A**) Group A 14 (25.4%) vs. Group C 10 (11.1%), *p* = 0.025; (**F**) Group B 17 (30.9%) vs. Group C 13 (14.4%), *p* = 0.018; (**G**) Group A 25 (45.4%) vs. Group C 26 (28.9%), *p* = 0.043; (**H**) Group A 36 (65.4%) vs. Group B 20 (36.4%), *p* = 0.002 ** and vs. Group C 37 (41.1%), *p* < 0.005 **; (**I**) Group A 31 (56.4%) vs. Group B 16 (29.1%), *p* = 0.004 ** and vs. Group C 32 (35.6%), *p* = 0.015 **. After Bonferroni correction significance was deemed retained by those comparisons with a *p*-value <0.016 only (highlighted by **).

**Table 1 viruses-13-00281-t001:** Clinical and virological characteristics of the study population.

Variable	Population
	(*n* = 200)
Median SARS-CoV-2 PCR Ct	
Group A, Ct ≤ 20.0	18.9 (17.9–19.5)
Group B, 20.0 < Ct ≤ 28.0	22.9 (22.0–25.2)
Group C, Ct > 28.0	34.0 (30.8–36.9)
Hospital admissions, n	127 (63.5%)
Symptomatic, n	180 (90.0%)
Signs and symptoms according to main systems, n	
Systemic inflammatory involvement	166 (83.0%)
Neurological involvement	73 (36.5%)
Gastroenterological involvement	35 (17.5%)
Respiratory involvement	131 (65.5%)
Time between symptoms onset and diagnostic swab collection, days *	5 (3–8)
Outcomes, n	
Complete recovery	127 (63.5%)
Sequelae	41 (20.5%)
Death	32 (16.0%)
Type of Sequelae, n	
Dyspnea	21 (12.5%)
Olfactory and/or gustatory dysfunction	13 (7.7%)
Chronic cough	7 (4.2%)
Others	6 (3.6%)
Worst oxygen support, n	
None	99 (49.5%)
Wall low-flow oxygen to reservoir	60 (30.0%)
CPAP	29 (14.5%)
Intubation	12 (6.0%)
Comorbidities, n	
None	82 (41.0%)
Hypertension	56 (28.0%)
COPD/asthma	33 (16.5%)
Overweight/obesity	27 (13.5%)
Active smoking	23 (11.5%)
Diabetes	19 (9.5%)
Cancer	17 (8.5%)
Others	46 (23.0%)
Disease severity, n	
Home isolation	73 (36.5%)
Hospital admission without oxygen support	26 (13.0%)
Hospital admission with low-flow oxygen to reservoir	39 (19.5%)
Hospital admission with CPAP	20 (10.0%)
Hospital admission with intubation	10 (5.0%)
Death	32 (16.0%)

* Asymptomatic patients excluded; others included 6 patients that reported overall 2 stroke-related hemiplegia, 2 reduction of vision, 1 recurrent infection, 1 hemolytic anemia requiring blood transfusions, 1 neuralgia, 1 chronic myalgia, 1 minor depression. CPAP, continuous positive airways pressure; COPD, chronic obstructive pulmonary disease.

**Table 2 viruses-13-00281-t002:** Univariate analysis, comparisons of demographic and clinical features between diagnostic SARS-CoV-2 Ct groups.

	A	B	C	p	η^2^	A vs. B *	A vs. C *	B vs. C *
	Ct ≤ 20	20 < Ct ≤ 28	Ct > 28					
	(n = 55)	(n = 55)	(n = 90)					
Age, years	64 (39–78)	57 (50–67)	52 (40–63)	0.017	0.396	0.381	0.011	0.025
Male sex, n	31 (56.4%)	32 (58.2%)	53 (58.9%)	0.956	-	0.848	0.766	0.933
Comorbidity, n								
None	15 (27.3%)	23 (41.8%)	49 (54.4%)	0.006	0.052	0.375	0.004	0.413
Hypertension	27 (49.1%)	13 (23.6%)	16 (17.8%)	<0.0005	0.077	<0.0005	0.009	0.999
COPD/asthma	12 (21.8%)	6 (10.9%)	15 (16.7%)	0.306	-	0.124	0.441	0.341
Overweight/obesity	8 (14.5%)	8 (14.5%)	11 (12.2%)	0.892	-	0.999	0.689	0.689
Active Smoking	12 (21.8%)	2 (3.6%)	9 (10.0%)	0.01	0.017	0.009	0.093	0.735
Diabetes	8 (14.5%)	6 (10.9%)	5 (5.6%)	0.186	-	0.569	0.067	0.239
Cancer	5 (9.1%)	5 (9.1%)	7 (7.8%)	0.947	-	0.999	0.781	0.781
Others	11 (20.0%)	19 (34.5%)	16 (17.8%)	0.08	-	0.164	0.515	0.028
Time from COVID-19 onset to swab collection, days	3 (2–5)	5 (3–9)	5 (3–10)	0.011	0.144	0.02	0.026	0.999
Number of signs and symptoms at diagnosis, n	4 (3–6)	3 (2–4)	3 (2–3)	0.007	0.058	0.037	0.008	0.977
Hospital admissions, n	41 (74.5%)	35 (63.6%)	51 (56.7%)	0.096	-	0.218	0.031	0.409
Worst oxygen support, n				0.495	-	0.923	0.28	0.377
None	24 (43.6%)	28 (50.9%)	47 (52.2%)					
Low-flow wall oxygen to reservoir	20 (36.4%)	11 (20.0%)	29 (32.2%)					
CPAP	7 (12.7%)	11 (20.0%)	11 (12.2%)					
Intubation	4 (7.3%)	5 (9.1%)	3 (3.3%)					
Outcomes, n				<0.0005	0.191	<0.0005	<0.0005	0.678
Complete recovery	16 (29.1%)	39 (70.9%)	72 (80.0%)					
Sequelae	19 (34.5%)	9 (16.4%)	13 (14.4%)					
Death	20 (36.4%)	7 (12.7%)	5 (5.6%)					
Disease severity, n				0.004	0.169	0.204	0.003	0.62
Home isolation	14 (25.4%)	20 (36.4%)	39 (43.3%)					
Hospital admission:								
Without oxygen support	10 (18.2%)	8 (14.5%)	8 (8.9%)					
With low-flow wall oxygen to reservoir	5 (9.1%)	8 (14.5%)	26 (28.9%)					
With CPAP	3 (5.4%)	7 (12.7%)	10 (11.1%)					
With intubation	3 (5.4%)	5 (9.1%)	2 (2.2%)					
Death	20 (36.4%)	7 (12.7%)	5 (5.6%)					
	A	B	C	p	η^2^	A vs. B *	A vs. C *	B vs. C *
	Ct ≤ 20	20 < Ct ≤ 28	Ct > 28					
	(n = 35)	(n = 48)	(n = 85)					
Outcomes among survivors, n				<0.0005	0.162	0.001	<0.0005	0.999
Complete recovery	16 (45.7%)	39 (81.2%)	72 (84.7%)					
Sequelae	19 (54.3%)	9 (18.8%)	13 (15.3%)					
Type of Sequelae, n								
Dyspnea	11 (57.9%)	1 (11.1%)	9 (69.2%)	0.022	0.004	0.067	0.999	0.024
O/G dysfunction	6 (31.6%)	4 (44.4%)	3 (23.1%)	0.579	-	0.677	0.704	0.376
Chronic cough	3 (15.8%)	3 (33.3%)	1 (7.7%)	0.294	-	0.352	0.264	0.264
Others	3 (15.8%)	1 (11.1%)	2 (15.4%)	0.945	-	0.998	0.976	0.999

COPD, chronic obstructive pulmonary disease; CPAP, continuous positive airways pressure; O/G dysfunction, olfactory and/or gustatory dysfunction. Eta-squared effect size for Kruskal–Wallis test was deemed as 0.01 < 0.06 small effect, 0.06 < 0.14 moderate effect, and ≥0.14 large effect. * Post hoc analysis, Bonferroni correction was applied in the pairwise comparisons when the *p*-value at Kruskal–Wallis H test was <0.05.

**Table 3 viruses-13-00281-t003:** Multivariate analysis for clinical outcomes, signs and symptoms at diagnosis, and long-term sequelae.

**COVID-19-Related Death (n 180) ***		
	**aOR (95CI)**	**p**
SARS-CoV-2 Ct	0.84 (0.72–0.97)	0.023
Age	1.25 (1.11–1.40)	<0.01
Sex	1.26 (0.21–7.55)	0.80
Time from COVID-19 onset to diagnostic swab	1.03 (0.91–1.18)	0.63
Number of comorbidities	2.44 (1.18–5.04)	0.016
Worst oxygen support None Low-flow wall oxygen-reservoir CPAP Intubation	– 2.01 (0.14–8.98) 2.14 (0.65–7.01) 2.38 (0.94–6.06)	- 0.61 0.34 0.068
**COVID-19 Severity (n 180) ***		
	**aOR (95CI)**	**p**
SARS-CoV-2 Ct	0.95 (0.91–0.99)	0.023
Age	1.08 (1.06–1.11)	<0.01
Sex	0.31 (0.17–0.58)	<0.01
Time from COVID-19 onset to diagnostic swab	1.04 (0.98–1.11)	0.21
Number of comorbidities	1.6 (1.29–2.03)	<0.01
**Number of signs and symptoms at diagnosis (n 180) ***		
	**β (95CI)**	**p**
SARS-CoV-2 Ct	−0.060 (−0.10; −0.018)	<0.01
Age	−0.013 (−0.32; 0.006)	0.17
Sex	−0.12 (−0.69; 0.44)	0.67
Number of comorbidities	0.24 (−0.013; 0.50)	0.063
Time from COVID-19 onset to diagnostic swab	−0.12 (−0.62; 0.51)	0.99
**6-month sequelae in survivors (n 149) ***		
	**aOR (95CI)**	**p**
SARS-CoV-2 Ct	0.90 (0.85–0.96)	<0.01
Age	1.02 (0.98–1.04)	0.30
Sex	0.59 (0.26–1.35)	0.21
Time from COVID-19 onset to diagnostic swab	0.98 (0.89–1.07)	0.63
Number of comorbidities	1.01 (0.68–1.48)	0.97
Worst oxygen support None Low-flow wall oxygen-reservoir CPAP Intubation	– 1.02 (0.37–2.82) 1.27 (0.37–4.38) 1.72 (0.41–7.29)	- 0.97 0.71 0.46

* Asymptomatic patients without a defined time from COVID-19 onset to diagnostic swab collection were not included in multivariate analysis. CPAP, continuous positive airways pressure.

## Data Availability

Raw data supporting the findings of this study are available from the corresponding author MT on request.

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
