# Peer review of "Diagnostic SARS-CoV-2 Cycle Threshold Value Predicts Disease Severity, Survival, and Six-Month Sequelae in COVID-19 Symptomatic Patients"

_viruses, 2021, doi:10.3390/v13020281_

Round 1

Reviewer 1 Report

It is my pleasure to review this paper

This paper reports the correlation between Ct value for the PCR test of SARS-CoV-2 and the results showed the viral load has significant bearing to the severity of the COVID-19 based on the set criteria to rate severity. Although there is no consensus of association between Ct value and severity of disease in COVID-19, this study may add on additional reference in literature for clinicians and researches to analyze

There are a few points that I would like to suggest to make this manuscript better for the readers

  1. For severity of disease, it was classified according to no admission, admission without oxygen, admission required oxygen, CPAP, intubation and death. While I agree using oxygen, CPAP, intubation and death may represent gradation of severity, it is not too clear about difference between no admission and admission without oxygen therapy. Maybe the author can further explain this in the methodology, the admission criteria and perhaps when oxygen will be given based on what clinical condition.
  2. The Ct value only is only checked on symptomatic cases but seldom on asymptomatic cases and this is the main drawback of this study as it is not uncommon to see cases with high Ct values but asymptomatic and become "super carriers" to spread the disease around. While this "super carriers" may be those immuno-compromised cases, they may not represent the mainstream of patients. I hope the author can address this with suitable references quoted in discussion to remind readers while interpreting the results.
  3. There is no description of the selection of Ct values for PCR of the SARS-CoV-2. When the sample of Nasopharyngeal and pharyngeal swab taken may have significant effect to the value. Usually it is high is the first week of infection and getting lowered. When there are a few PCR test for each patient after admission, which value to take for analysis or average value of Ct within a time frame should be clearly described.
  4. There are a few factors that may predict severity of the COVID-19 infection. Old age and comorbidity (having chronic illnesses) seem to be the most significant factors that we observed and has been shown in this study. Even though Ct value is also statistically significant, author can comment on its significance in clinical management eg, anti-viral medications to start based on the values of Ct of PCR and the response.
  5. Lastly, in this study no correlation is noted between Ct value and incidence of smell and taste loss in COVID-19. Although no rating scale for smell loss and taste loss had been used to analyze their severity and Ct value relationship, author can still attempt to explain the possible reasons for the former finding in this study.

I hope the author can revise the manuscript based on the above suggestions and other reviewers' suggestion to better the content of this article before publication in "Viruses". I appreciated the effort from all coauthors and the team to contribute in this study while the country was still in critical situation under COVID-19. Please stay safe and stay strong!!

Author Response

This paper reports the correlation between Ct value for the PCR test of SARS-CoV-2 and the results showed the viral load has significant bearing to the severity of the COVID-19 based on the set criteria to rate severity. Although there is no consensus of association between Ct value and severity of disease in COVID-19, this study may add on additional reference in literature for clinicians and researches to analyse

AR: We are grateful to Reviewer1 for her/his time spent at improving our manuscript and for her/his concluding appreciations and encouragement! All the modifications to the manuscript have been highlighted in red.

There are a few points that I would like to suggest to make this manuscript better for the readers

1. For severity of disease, it was classified according to no admission, admission without oxygen, admission required oxygen, CPAP, intubation and death. While I agree using oxygen, CPAP, intubation and death may represent gradation of severity, it is not too clear about difference between no admission and admission without oxygen therapy. Maybe the author can further explain this in the methodology, the admission criteria and perhaps when oxygen will be given based on what clinical condition.

AR: We agree with the fact that the less a sample of 200 patients is stratified into subgroups the better it is; still, we decided not to consider as a whole “not hospitalized patients and those hospitalized not in need of oxygen” for at least two reasons. First, data on not hospitalized patients are very scarce and most of the other studies on the theme report data on hospitalized patients only; therefore, having 76 not hospitalized patients to be considered as a separate category was a small but initial possibility to say something also in this under-represented category. Secondly, in terms of disease severity, we did not feel confident in summing up these two groups since there is a significant difference between them in terms of several variables that we were not able to take into account in our study (such as treatments –although in March there were no guidelines no approaches that later on proved an impactful effect upon COVID-19-, and general clinical evaluations considered by physicians at the time of health-care access).

As for the admission criteria and oxygen administration, the sampling frame and the sampled patients belong to a very large area in Piedmont. In March, many hospitals sent nasal swabs to our laboratory to be analysed from Torino city, but also from surrounding areas, counties and districts (we have better specified it now in page 3, lines 2-3). Therefore, the clinical decision making and management of these patients is very heterogeneous as admitted to different hospitals, wards and evaluated by different emergency departments and physicians. There was undoubtedly heterogeneity in hospitalization criteria (subjective evaluation of physicians, availability of beds, local guidelines) and likely less in oxygen administration (roughly if SpO2<94% in previously healthy subjects and if SpO2<92% in patients with COPD). Indeed, reporting strict criteria for hospital admission or oxygen administration in our methods section would be a false statement. We have retrospectively collected and classified these as ordinal or binary variables according to the self-reporting of patients and according to the clinical records from the on-line platform (which indeed reports about type of oxygen support and hospitalisation, but not about the parameters used to make these clinical decisions). As we cannot specify further, we have highlighted this limitation in the discussion (see page 10, fifth paragraph: “Furthermore, the heterogeneity in the clinical management of patients coming from such a large area of northwestern Italy may limit the homogeneity of certain subgroups as oxygen administration as well as hospitalization varied and relied on several factors we were not able to consider (as ex. physicians evaluations, local guidelines, availability of intensive care unit beds”).

2. The Ct value only is only checked on symptomatic cases but seldom on asymptomatic cases and this is the main drawback of this study as it is not uncommon to see cases with high Ct values but asymptomatic and become "super carriers" to spread the disease around. While this "super carriers" may be those immuno-compromised cases, they may not represent the mainstream of patients. I hope the author can address this with suitable references quoted in discussion to remind readers while interpreting the results.

AR: We agree with Reviewer1; despite the 20 asymptomatic patients (10%), we have focused the conclusions of our analysis in symptomatic patients only starting from the title of the paper that states that the considered clinical outcomes are associated with the diagnostic SARS-CoV-2 cycle threshold in Symptomatic patients. Indeed, also multivariate models did not include those 20 patients as for them it was impossible to calculate the time gap between symptoms onset and swab collection, and we have specified this in the notes to Table 3 (page 7). We added in the discussion of the limitations of our work a paragraph to better clarify that our results can be applied to symptomatic subjects only and that there are still knowledge gap regarding the association between the initial amount of virus and the eventual clinical evolution of the disease (last paragr of page 10 and following in page 11: “As we sampled only a 10% of asymptomatic infections, further studies are warranted to better clarify why the relationship between Ct values (or the amount of nasal-pharyngeal virus) and clinical outcomes seems to follow a different pattern in subjects developing asymptomatic SARS-CoV-2 infection. Indeed, several cohorts described no difference in Ct values between symptomatic and asymptomatic patients [6,25,41], so that genetic, immunological, analytical or viral factors may interpose in the relationship we have observed.”). Furthermore, we specified “symptomatic patients” whenever it was required (see page 11, fourth line or page 11, beginning of Conclusions).

3. There is no description of the selection of Ct values for PCR of the SARS-CoV-2. When the sample of Nasopharyngeal and pharyngeal swab taken may have significant effect to the value. Usually it is high is the first week of infection and getting lowered. When there are a few PCR test for each patient after admission, which value to take for analysis or average value of Ct within a time frame should be clearly described.

AR: we have considered for our analysis only the cycle threshold value of RdRp gene detected in the first swab that made the diagnosis of COVID-19 for each included patients, but since the term “diagnostic” may ingenerate confusion and it was not clear, we have better specified now the point raised by the reviewer (see page 3, beginning of second parag.). We have also reported the time between the onset of COVID-19 signs and symptoms and the time of swab collection (in the overall sample at a median time of 5 days –see Table 1, page 4, fifth variable-, and further stratified according to the Ct groups, see page 5, third line of the paragr. and Table 2, fourth variable) to correct for viral dynamics and potential delays that may affect the amount of detected virus (and therefore the classification of patients in Ct groups), as the reviewer underlined. Multivariate analysis always included among the covariates this temporal variable as we totally agree with Reviewer1 about its fundamental importance when analyzing such a relationship (see Table 3). We have also commented on the importance of considering temporal trends and viral kinetics in similar analysis, underlying the fact that only few studies in the field, including ours, reported the time frame of the swab collection (see page 9, third par. of the discussion “Considering viral kinetics and immune-pathology dynamics [5,28,30,34], SARS-CoV-2 viral load should be adjusted for this temporal variables to avoid misclassification of patients…”).

4. There are a few factors that may predict severity of the COVID-19 infection. Old age and comorbidity (having chronic illnesses) seem to be the most significant factors that we observed and has been shown in this study. Even though Ct value is also statistically significant, author can comment on its significance in clinical management eg, anti-viral medications to start based on the values of Ct of PCR and the response.

AR: we have now expanded the discussion on potential clinical significance and usefulness of Ct value (see page 11, third paragr.; page 11, last sentence of the discussion and second paragr. of conclusions).

5. Lastly, in this study no correlation is noted between Ct value and incidence of smell and taste loss in COVID-19. Although no rating scale for smell loss and taste loss had been used to analyze their severity and Ct value relationship, author can still attempt to explain the possible reasons for the former finding in this study.

AR: We have not designed the study to primarily assess an association between Ct value and O/G dysfunction and therefore, after random sampling, we ended up having only 69 patients with O/G dysfunction at diagnosis (prevalence of 34.5% that is lower than that reported from a recent meta-analysis on 10 cohorts, 52-43%). We have modified the comment on this part in the discussion to acknowledge better that the discrepancy of our negative finding compared to recent findings reporting an association should not be surprising, including primarily the statistical limitations and including potential biological explanations (see page 10, third paragraph). In this regard, we also added and commented on the overall prevalence of sequelae at 6 months, as a recent paper reported the data at the beginning of January (see page 10, fourth paragr.)

I hope the author can revise the manuscript based on the above suggestions and other reviewers' suggestion to better the content of this article before publication in "Viruses". I appreciated the effort from all coauthors and the team to contribute in this study while the country was still in critical situation under COVID-19. Please stay safe and stay strong!!

Thanks again to Reviewer 1 for her/his comments and time spent at improving our manuscript!

Reviewer 2 Report

The manuscript “Diagnostic SARS-CoV-2 Cycle Threshold Value Predicts Disease Severity, Survival and 6-month Sequelae in COVID-19 Symptomatic Patients” by Trunfio and colleagues evaluates 3 tiers of RdRP Ct values (A≤20.0, 20.0<B≤ 28.0, C>28.0) in 200 patients from the Italian region of the Piedmont (both hospitalized patients and outpatient), with respect to comorbidities, symptoms, disease severity, outcomes and long-term sequelae. The study design, choice of clinical variables, cutoffs and statistical tools (pending clarification on multiple testing correction) are reasonable and adequately justified by the authors. The authors’ scholarly introduction identifies current gaps in the literature; however, some of the listed limitations are ultimately shared by their own work. While the authors’ results do for the most part support their conclusions and the limitations are understandable, some of the language used in the introduction (such as “no pathogen-specific prognostic biomarkers are yet readily available for SARS-CoV-2”, “it is a current matter of debate whether SARS-CoV-2 viral load is an impactful factor in determining disease outcomes”, or “heterogeneous recruitment criteria [have hampered] a final firm conclusion on the relationship between initial nasopharyngeal viral load and individual prognosis”) set up the reader for the expectation that this study will resolve these issues - a hard bar for any single study to meet, especially one that is moderate in size. Ultimately, this study represents a valuable addition to the growing body of literature surrounding the issue of the prognostic significance of SARS-CoV-2 Ct values and viral loads in a well-defined population from Italy. The most notable features of this study include the long follow up period (6 months), the inclusion of a study sample with diverse presentations and outcomes, the cross-referencing of a hospital and contact-tracing platform, and a well-designed and simple but still comprehensive retrospective evaluation. The main limitation is the study size of n=200. 

Major points: 

1.       The original dataset contained 1995 samples (1138 with Ct > 28 and 857 with Ct < 28), yet only 230 (randomly selected) were included in the study, of which 30 refused to participate or did not answer, leaving the final ‘n’ at 200 (the provided power calculation suggested a study size of 225). While several of the authors’ findings reportedly reach statistical significance (see #2), others do not, especially when dealing with stratified parameters. As the authors themselves point out, cohort size is often a limitation in many studies seeking to determine the relationship between viral load and disease severity. This seems like a missed opportunity given the size of the available starting pool.  

2.       Please explain in greater detail which corrections are applied for multiple testing. Where possible, please list the precise (corrected) p-value to a greater number of significant digits, rather than listing it as “<0.01”. 

3.       The trends for group B are at times not intermediate between group A and C. This is unexpected and would benefit from being addressed in the discussion. 

Minor points: 

1.       Journal directions are retained in the ‘Materials and Methods’ section (first 3 paragraphs) 

2.       Discussion of ref. 33 (253-260) is unnecessary and speculative. It could be removed. The same is true of the discussion on oxygen support in lines 301-305. It is unclear how this relates to the rest of the work. (Admittedly, these may have been requested by a previous reviewer.) 

3.       It would help with readability if A, B and C in tables and figures had labels such as A (Ct ≤20.0), B (20.0<Ct≤ 28.0) and C (Ct>28.0). 

4.       The ‘Conclusions’ section should directly address the author’s evaluation of the impact of their own results on the field. It currently consists of general comments regarding the current status quo and important open questions. 

As a side note, I appreciate that the authors refer to their variable as cycle threshold (Ct) value. Many publications use Ct values (semi-quantitative) but casually discuss them as viral loads (quantitative), which is incorrect. Ct values, especially when used to compare populations evaluated with the same diagnostic platform, as is the case here, can be valuable and serve as cumulative evidence to justify the transformation of qualitative to quantitative assays and/or the creation of guidelines for interpreting Ct values and viral loads. The authors’ use of Ct value throughout the manuscript (including the title) is refreshing and indicates rigor and honesty.

Author Response

The manuscript “Diagnostic SARS-CoV-2 Cycle Threshold Value Predicts Disease Severity, Survival and 6-month Sequelae in COVID-19 Symptomatic Patients” by Trunfio and colleagues evaluates 3 tiers of RdRP Ct values (A≤20.0, 20.0<B≤ 28.0, C>28.0) in 200 patients from the Italian region of the Piedmont (both hospitalized patients and outpatient), with respect to comorbidities, symptoms, disease severity, outcomes and long-term sequelae. The study design, choice of clinical variables, cutoffs and statistical tools (pending clarification on multiple testing correction) are reasonable and adequately justified by the authors. The authors’ scholarly introduction identifies current gaps in the literature; however, some of the listed limitations are ultimately shared by their own work. While the authors’ results do for the most part support their conclusions and the limitations are understandable, some of the language used in the introduction (such as “no pathogen-specific prognostic biomarkers are yet readily available for SARS-CoV-2”, “it is a current matter of debate whether SARS-CoV-2 viral load is an impactful factor in determining disease outcomes”, or “heterogeneous recruitment criteria [have hampered] a final firm conclusion on the relationship between initial nasopharyngeal viral load and individual prognosis”) set up the reader for the expectation that this study will resolve these issues - a hard bar for any single study to meet, especially one that is moderate in size. Ultimately, this study represents a valuable addition to the growing body of literature surrounding the issue of the prognostic significance of SARS-CoV-2 Ct values and viral loads in a well-defined population from Italy. The most notable features of this study include the long follow up period (6 months), the inclusion of a study sample with diverse presentations and outcomes, the cross-referencing of a hospital and contact-tracing platform, and a well-designed and simple but still comprehensive retrospective evaluation. The main limitation is the study size of n=200. 

AR: We are grateful to Reviewer2 for her/his time spent at improving our manuscript and for appreciating that the long follow-up is a plus rather than a limitation of this study.

Major points: 

1. The original dataset contained 1995 samples (1138 with Ct > 28 and 857 with Ct < 28), yet only 230 (randomly selected) were included in the study, of which 30 refused to participate or did not answer, leaving the final ‘n’ at 200 (the provided power calculation suggested a study size of 225). While several of the authors’ findings reportedly reach statistical significance (see #2), others do not, especially when dealing with stratified parameters. As the authors themselves point out, cohort size is often a limitation in many studies seeking to determine the relationship between viral load and disease severity. This seems like a missed opportunity given the size of the available starting pool. 

AR: We agree with regret with Reviewer2 about this missed opportunity. Unfortunately, the phone surveys especially required a massive effort in terms of time to be carried on. All the professionals involved in data collection, analyses etc are physicians and microbiologists (that were and are still working in COVID-19 wards), as we do not have data manager or other dedicated figures. We had to make a compromise between energy and efforts dedicated to clinical practice and clinical research, so that we set the sample size based on the primary outcome “difference in mortality”. We hope this work may be still worthy, as it still relies on a planned sample size (as not all studies published on the same theme) adding further evidence to the growing one and reports as first the description of a potential association between Ct values at diagnosis and 6-months sequelae.

2. Please explain in greater detail which corrections are applied for multiple testing. Where possible, please list the precise (corrected) p-value to a greater number of significant digits, rather than listing it as “<0.01”.

AR: Thank you for pointing out this issue. Originally, we have not applied any multiple testing corrections. We thought that the primary analysis (CT and COVID-19 severity) did not require adjustment for multiple testing since it is a bivariate correlation that is the core of our work; multivariate analysis then adjusted for the variable we identified and reduced the risk of "by chance" results in our dataset. After discussing further, we have decided to balance the suggestion of the Reviewer2 with the risk of increasing type II errors. We have now reported in Table 2 and in the text (see page 7, second paragraph) the eta-squared effect size for the three-group comparisons with a significant p-value or trend to better help readers in interpreting the significance and magnitude of the observed “significant differences”. All the eta-squared resulted of a moderate or large effect size; only the eta-squared of the comparisons among the three groups in the difference in olfactory/gustatory dysfunction prevalence at 6 months, number of signs and symptoms at diagnosis, “having no comorbidity” and involvements in specific systems resulted in a small effect size despite a significant p-value.

Furthermore, we have applied Bonferroni correction within the post-hoc one-by-one comparisons (A vs B etc) in Table 2 and Figure 1 footnotes (and we have specified it in the statistical analysis section, see page 3-4, highlighted in red); Bonferroni did not lead to a significant loss of “significance”.

We have also acknowledged that the study is underpowered for our secondary outcomes (specific signs and symptoms, specific difference in sequelae types etc) in the discussion section (see page 9, third paragr of the discussion and page 10, fourth-fifth lines of the third paragr.).

As for p values we included all for decimal digits in the figures, tables and text (<0.0005 was used when the SPSS output reported 0.000), as highlighted by modified p-values in red color.

3. The trends for group B are at times not intermediate between group A and C. This is unexpected and would benefit from being addressed in the discussion.

AR: We have tried to explain this finding in the discussion section (see page 9, third paragr. of the discussion section); still, we believe that the core analysis and messages of our work is represented by the multivariate analyses that support our conclusions and may have overcome potential arbitrariness of the Ct value cut-off of 20 as well as potential effects of differences in time to swab collection from the onset of COVID-19 that in univariate analysis may have lead us to classify as B some patients actually belonging to group A if tested more precociously (there is a 2 days delays in testing group B compared to A which is significant and it was possible to correct for it only at multivariate analysis).

Minor points: 

  1. Journal directions are retained in the ‘Materials and Methods’ section (first 3 paragraphs)

AR: apology for the mistake, we have removed the retained paragraphs.

  1. Discussion of ref. 33 (253-260) is unnecessary and speculative. It could be removed. The same is true of the discussion on oxygen support in lines 301-305. It is unclear how this relates to the rest of the work. (Admittedly, these may have been requested by a previous reviewer.)

AR: thank you, we have removed both the sentences.

  1. It would help with readability if A, B and C in tables and figures had labels such as A (Ct ≤20.0), B (20.0<Ct≤ 28.0) and C (Ct>28.0).

AR: We have corrected tables and figures as suggested.

  1. The ‘Conclusions’ section should directly address the author’s evaluation of the impact of their own results on the field. It currently consists of general comments regarding the current status quo and important open questions.

AR: We have modified as suggested, moving the previous conclusions at the end of the discussion and writing the new conclusions as below (page 11): “Among symptomatic hospitalized and not hospitalized patients, we demonstrated an association of the Ct value detected in nasal-pharyngeal swabs collected within the first week from COVID-19 onset with COVID-19-related deaths, disease severity, and number of signs and symptoms at diagnosis as well as, to the best of our knowledge, for the first time with the persistence of sequelae at 6 months. These relationships were retained even after adjusting for other relevant parameters already expected to affect the amount of virus at the beginning of the infection. Further confirmation of our observations and the identification of reliable standardized Ct cut-offs, together with other already better known determinants of COVID-19 severity, could lead to timely differentiated paths in terms of clinical monitoring and management of patients when accessing health facilities as well as tailored follow-up in terms of duration and type of required assessments. Whenever possible, randomized controlled trials and other studies aiming at assessing efficacy or effectiveness of drugs in COVID-19 should report also on the amount of initial virus or its proxy to correct for another parameter that is emerging as relevant in the outcomes of SARS-CoV-2 infection”

As a side note, I appreciate that the authors refer to their variable as cycle threshold (Ct) value. Many publications use Ct values (semi-quantitative) but casually discuss them as viral loads (quantitative), which is incorrect. Ct values, especially when used to compare populations evaluated with the same diagnostic platform, as is the case here, can be valuable and serve as cumulative evidence to justify the transformation of qualitative to quantitative assays and/or the creation of guidelines for interpreting Ct values and viral loads. The authors’ use of Ct value throughout the manuscript (including the title) is refreshing and indicates rigor and honesty.

Thank you, once again to Reviewer2 for his/her time and comments to improve our manuscript and his/her final appreciation.

Reviewer 3 Report

The manuscript presents very interesting results, and contributes to demonstrate the correlation between nasopharyngeal viral load at diagnosis and development and outcome of COVID-19, also on the long-term. The results are presented in a very clear and linear way, and the discussion is well structured and consistently appropriate to the results obtained.

I submit only one aspect to your attention. RT-PCR for SARS-CoV-2 genomic RNA detection makes use of many different kits, processed on different instruments, not always directed on the same number and/or type of viral targets. Although RpRd is constantly analyzed, the sensitivity in its detection by different manufacturers can still vary. For this reason, a cut-off defined for a particular kit, will not necessarily be applicable to others. If the definition of "high viral load" identified in this study is to be used as a prognostic factor, it will need to be standardized to take this into account.

Please check the initial part of the materials and methods.

Author Response

The manuscript presents very interesting results, and contributes to demonstrate the correlation between nasopharyngeal viral load at diagnosis and development and outcome of COVID-19, also on the long-term. The results are presented in a very clear and linear way, and the discussion is well structured and consistently appropriate to the results obtained.

AR: We are grateful to Reviewer3 for her/his time spent at improving our manuscript and for her/his appreciation of our work.

I submit only one aspect to your attention. RT-PCR for SARS-CoV-2 genomic RNA detection makes use of many different kits, processed on different instruments, not always directed on the same number and/or type of viral targets. Although RpRd is constantly analyzed, the sensitivity in its detection by different manufacturers can still vary. For this reason, a cut-off defined for a particular kit, will not necessarily be applicable to others. If the definition of "high viral load" identified in this study is to be used as a prognostic factor, it will need to be standardized to take this into account.

AR: We agree with the reviewer. Indeed, we have acknowledged the limitations and variability in Ct assessment and we extended the discussion on the point raised by the reviewer: page 9, last sentence of the third paragr. of conclusions (“Further studies addressing the best, standardized and most reliable assay-based and gene-based Ct value cut-offs should be performed to support a potential clinical and easily interpretable application of Ct values in routine practice.”) and page 11, fourth paragr. (“The interpretation of a single Ct value should still be performed cautiously as it may be affected by sample collection, analyzed gene, adopted assay, and analytic limits [44]; therefore, reference Ct cut-offs should be validated and standardized by genes and assays before any application in clinical practice”). The aim of our study was exploratory in its nature. We needed a cut off (partially arbitrary as explained in the methods section) to create groups, but we did not conclude suggesting to use it to stratify patients in terms of risk in routine practice as we did not perform a prognostic/diagnostic accuracy study.

Please check the initial part of the materials and methods.

AR: apologies, we removed the retained part at the beginning of materials and methods.

Thanks to Reviewer3 once again for her/his time spent at improving our manuscript.

Reviewer 4 Report

Trunfio and colleagues present their work on the association of SARS-CoV-2 cycle threshold values with short and long term COVID-19 outcomes and report that lower Ct values are associated with disease severity, death and long term sequalae. This work comes to add to a growing amount of data from other studies that show the importance of Ct values on disease prognosis.

The authors should acknowledge previous work on the subject and discuss who their work adds to existing knowledge.

  • Ann Am Thorac Soc. 2020 Oct 29. doi: 10.1513/AnnalsATS.202008-931RL.
  • PLoS One. 2020 Nov 17;15(11):e0242399. doi: 10.1371/journal.pone.0242399. eCollection 2020.

The categorization of Ct values is arbitrary. The authors should perform sensitivity analysis dividing Ct values in tertiles but also using Ct value as a continuous variable to show the robustness of their results.

Author Response

Trunfio and colleagues present their work on the association of SARS-CoV-2 cycle threshold values with short and long term COVID-19 outcomes and report that lower Ct values are associated with disease severity, death and long term sequalae. This work comes to add to a growing amount of data from other studies that show the importance of Ct values on disease prognosis.

AR: We are grateful to Reviewer4 for her/his time spent at improving our manuscript.

The authors should acknowledge previous work on the subject and discuss who their work adds to existing knowledge.

  • Ann Am Thorac Soc. 2020 Oct 29. doi: 10.1513/AnnalsATS.202008-931RL.
  • PLoS One. 2020 Nov 17;15(11):e0242399. doi: 10.1371/journal.pone.0242399. eCollection 2020.

AR: we have included among the references of previous works on the same topic those suggested by reviewers 4 (now references 35 and 36). We have also specifically commented on the second suggested paper in the discussion (page 11, fourth paragr. “Interestingly, it has been recently described how the variation in time of sequential Ct values from nasopharynx of infected subjects sensibly predicts the changes in clinical status of COVID-19 cases [35].”), as it is the single one to the best of our knowledge to have shown how to use dynamics changes of Ct values as a prognostic clinical factor.

We have also commented on what our study add to previous published evidence (page 9, paragraphs 2-4 of discussion; page 11, first paragraph of the new conclusions) and extended a comparison of our data with those on sequelae prevalence at 6 months published at the beginning of this month (see page 10, fourth paragraph).

The categorization of Ct values is arbitrary. The authors should perform sensitivity analysis dividing Ct values in tertiles but also using Ct value as a continuous variable to show the robustness of their results

AR: We agree with Reviewer4 that the Ct categorization is arbitrary, but using tertiles would lead to use as cut-off 22.0 and 30.7, which are not extremely far from those chosen and would be arbitrary and self-referencing as well since they would depend on the characteristics of the study sample. More importantly, we have preferred to choose cut-offs based on biological hypotheses derived from previous data on viral culture infectivity and transmission risks, as reported in the methods. We know that Ct cut-offs should be based also on the used assay and analysed gene and ours are different from those used in the referenced studies aforementioned, but it has been shown also a difference of about ±1.0 in the reported Ct values of different genes and assays (see as ex. Uhteg K et al, Comparing the analytical performance of three SARS-CoV-2 molecular diagnostic assays, J Clin Virol, 2020). Therefore, we still believe that Ct cut-offs based on plausible biological thresholds could help further studies in confirming our results and, in the light of potential application into clinical practice, in using cut-offs less dependent on the analyzed sample. Taking into account the already vast amount of analyses performed and reported in the paper, we fear that further sub-analyses using Ct values as a continuous variable would just heavy the readability of the text without adding much as Ct values were indeed used as a linear continuous variable in all multivariate analyses.

Thanks to Reviewer4 once again for her/his time spent at improving our manuscript.

Round 2

Reviewer 2 Report

The authors of “Diagnostic SARS-CoV-2 Cycle Threshold Value Predicts Disease Severity, Survival and 6-month Sequelae in COVID-19 Symptomatic Patients” have addressed my comments, questions and suggestions in a satisfactory, detailed and timely fashion. I appreciate the additional context regarding the final sample size of the study and the substantially revamped statistics. Whereas the original version of the manuscript did not provide enough details on how the statistics were performed, in this revised version the statistics have become a strength of the paper. 

Addressing point by point: 

Original major point #1: Thank you for your transparency and openness. I understand the circumstances and commend you for your commitment to patient care and research in these trying times. It is my hope that you will have the resources in the future to make full use of the valuable information you have collected. Having said that, I agree that the current manuscript does a good job of evaluating the planned primary outcome and is a good addition to the current literature. 

Original major point #2: Thank you for putting in the time and effort to provide a substantive revision to the statistics in your manuscript. I agree with the authors that the multiple correction testing is less relevant to the bivariate correlation in the primary analysis. My remark was primarily concerned with the post-hoc testing in Figure 1, which has been addressed by applying a Bonferroni correction. I was also hoping for increased transparency and details regarding the methods used, something the authors have also accomplished. The inclusion of the effect size to the primary analysis is a valuable addition to the manuscript and improves its quality. 

Original major point #3: Adequately addressed. 

Original minor points #1-3: Adequately addressed. 

Original minor point #4: Thank you for writing a new conclusion that captures and highlights the contributions of your work. 

Author Response

Thanks to the Reviewer once again for his/her time and efforts to improve our manuscript.

there are no major or minor issues to be addressed.

Reviewer 4 Report

Thank you for addressing my comments. 

Author Response

Thanks to the Reviewer once again for his/her time and efforts to improve our manuscript.

There are no major or minor issues to be addressed.